# Progress in the Utilization of Coal Fly Ash by Conversion to Zeolites with Green Energy Applications

**DOI:** 10.3390/ma13092014

**Published:** 2020-04-25

**Authors:** Silviya Boycheva, Denitza Zgureva, Katerina Lazarova, Tsvetanka Babeva, Cyril Popov, Hristina Lazarova, Margarita Popova

**Affiliations:** 1Department of Thermal and Nuclear Power Engineering, Technical University of Sofia, 8 Kl. Ohridsky Blvd, 1000 Sofia, Bulgaria; 2College of Energy and Electronics, Technical University of Sofia, 8 Kl. Ohridsky Blvd, 1000 Sofia, Bulgaria; dzgureva@gmail.com; 3Institute of Optical Materials and Technologies ‘‘Acad. J. Malinowski’’, Bulgarian Academy of Sciences, Acad. G. Bonchev Str., bl.109, 1113 Sofia, Bulgaria; klazarova@iomt.bas.bg; 4Institute of Nanostructure Technologies and Analytics (INA), University of Kassel, Heinrich-Plett-Str. 40, 34132 Kassel, Germany; popov@ina.uni-kassel.de; 5Institute of Organic Chemistry with Centre of Phytochemistry, Bulgarian Academy of Sciences, bl.9, 1113 Sofia, Bulgaria; lazarova@orgchm.bas.bg (H.L.); mpopova@orgchm.bas.bg (M.P.)

**Keywords:** fly ash utilization, fly ash zeolites, catalytic oxidation of VOCs, CO_2_ adsorption, fly ash zeolite films

## Abstract

Fly ash (FA) from lignite coal combusted in different Thermal Power Plants (TPPs) was used for the synthesis of zeolites (FAZs) of the Na-X type by alkaline activation via three laboratory procedures. FAZs were characterized with respect to their morphology, phase composition and surface properties, which predetermine their suitability for applications as catalysts and adsorbents. FAZs were subsequently modified with metal oxides (CuO) to improve their catalytic properties. The catalytic activity of non-modified and CuO-modified FAZs in the total oxidation of volatile organic compounds was investigated. FAZs were studied for their potential to retain CO_2_, as their favorable surface characteristics and the presence of iron oxides make them suitable for carbon capture technologies. Thin films of FAZs were deposited by in situ crystallization, and investigated for their morphology and optical sensitivity when exposed to pollutants in the gas phase, e.g., acetone. This study contributes to the development of novel technological solutions for the smart and valuable utilization of FA in the context of the circular economy and green energy production.

## 1. Introduction

Coal remains the main energy source on a global scale. The combustion of coal in Thermal Power Plants (TPPs) generates numerous atmospheric pollutants, including fine particulate matters. The discharge of ash particles (fly ash, FA) into the atmosphere is avoided by their separation from flue gases in dust collectors. However, the subsequent landfilling of FA poses risks to the environment due to changes in the chemistry of the ground water, acidification of soils, and accumulation of potentially toxic elements [1]. The current increased responsibility for environmental protection requires the development of a new generation of coal-fired TPPs via the implementation of near zero-emission and waste recovery technologies. In this context, FA from coal is considered a raw material rather than a waste, and in industrial settings it is utilized in applications such as concrete production, road base construction, and soil amendment [2,3]. However, these applications reduce the amounts of solid wastes, but do not exploit fully the potential uses of this abundant resource [4,5]. Therefore, smart and more practically valuable approaches for the utilization of FA have been under development in recent years. For example, fly ash particles have been investigated as fillers in polymers [6], in the processing of architectural ceramics [7], and as a source of other valuable materials [8,9]. The conversion of FA into zeolites has potentially the greatest environmental benefits due to the broad application of these materials in water-, air- and soil-cleaning technologies. 

Coal fly ash zeolites (FAZs) are intensively investigated regarding their potential applications in waste water remediation, flue gas purification, and for catalytic degradation of air contaminants [10,11,12]. FAZs are typically characterized by smaller specific surface area when compared to their pure synthetic counterparts or natural minerals. Meanwhile, these materials possess a number of advantages that make them valuable for the practice. Coal fly ash zeolites are distinguished in their significant content of iron oxides (γ-Fe_2_O_3_, α-Fe_2_O_3_, γ-Fe_3_O_4_) and doping elements (e.g., Cu, Co, Mn, V, W) transferred from raw FA. The presence of spinel iron oxides gives FAZs magnetic properties, whereas the uniform distributions of Fe^2+^/Fe^3+^ centers and metal nanoparticles are a prerequisite for their excellent catalytic activity [12]. In addition, the formation of Brønsted acidic sites on the porous aluminosilicate structural framework of FAZs stipulates their performance as efficient catalytic systems, integrating a carrier and active sites. This allows for the application of FAZs in the oxidative degradation of volatile organic compounds (VOCs). VOCs are harmful atmospheric pollutants generated in manufacturing and the use of paints, varnishes and solvents; they are emitted as direct or secondary products from transportation and combustion of petrol fuels, biomass, and wastes [13]. Noble metals (Pt and Pd) are among the most employed catalysts for the thermal degradation of VOCs, but since they are expensive and sensitive to poisoning, their low-cost alternatives are intensively studied [14]. Suitable substitutes of the expensive noble metal catalysts are the materials containing transition metals (Co, Cu, Fe, Cr, etc.), whose fine dispersion onto porous supports with high surface area creates difficulties to obtaining efficient catalytically active systems [15]. FAZs are porous matrices with distributed active centers, and can be considered as viable alternatives to catalytic systems for the oxidation of VOCs. The efficiency of FAZ catalysts can be further improved via post-synthesis modification with metal oxides. 

FAZs are also distinguished by their mixed micro–mesoporous structure, facilitating mass transport phenomena through the material, which is beneficial for their adsorption and catalytic applications [16]. Due to their strong surface unsaturation, FAZs have adsorption potential comparable to their pure synthetic analogues and outperform them in a more energy-efficient desorption process [17]. The potential use of FAZs as adsorbents in carbon capture technologies offers a technological solution for closed-cycle environmental protection in coal-fired TPPs. For these purposes, FA can be converted into CO_2_ adsorbents, thus reducing, at the same time, the environmental impact of both solid waste and greenhouse gases generated by combustion processes. Recent studies on the CO_2_ adsorption capacities of FAZs have revealed promising results [17,18]. It has been established that among the zeolites, faujasite (FAU) and linde (LTA) types possess the greatest potential for CO_2_ capture due to the favorable combination of their thermal and mechanical stability, and their beneficial structural frameworks with appropriate pore sizes [19]. CO_2_ uptake onto zeolites can be improved further, not only by the optimization of operational conditions, but also by modifying their microporous structure to create hierarchical micro-mesoporosity. In addition, lowering Si to Al ratio in the zeolite composition will increase the surface heterogeneity and strengthen the electrostatic field inside the pores, enhancing the adsorption of CO_2_ [20]. The type of the compensating cations in the zeolite framework also impacts strongly on their CO_2_ adsorption potential [21]. A zeolite of FAU type (Na-X) has been selectively synthesized from high-silica FA by a double-stage fusion–hydrothermal activation, reaching up to 62% crystallinity [22]. Multistage fusion–hydrothermal synthesis preceded by FA calcinations and acidic treatment was performed by Volli and Purkait [23] to obtain single-phase zeolite Na-X. However, middling specific surface values of about 160 m^2^/g FAZ have been reported. The optimization of procedures for conversion of FA to zeolite Na-X, which is one of the zeolites with huge practical importance, regarding the yield of the main phase, as well as the economy and energy efficiency of the process, is of significant practical concern. It has been observed that FAZ of Na-X type can be also obtained by applying low- or zero-energy synthesis methods, such as a single stage of hydrothermal activation and long-term crystallization at ambient conditions [24,25]. However, the hydrothermal alkaline treatment of FA results in the low crystallinity of Na-X and in the crystallization of accompanied phases in the zeolite products [24]. Alkaline aging of FA at ambient conditions leads to a significant yield of the zeolite Na-X formation after long-term incubation of the reaction mixtures for several months. Thus, the need for external energy is eliminated and the equipment required is simplified [25]. Despite the technological drawbacks of the hydrothermal and long-term aging processes, the benefits of these processes should be considered as viable ways of processing FA into zeolite Na-X because of the low cost and technological simplicity of the processes. Bearing in mind that the coal ash wastes are deposited for a long period, the duration of their processing is not a meaningful critical issue.

FAZs are also considered as optional materials in CO_2_ capture due to their low-cost processing [18]. Moreover, these materials are promising adsorbents due to their mixed porosity, and high surface heterogeneity. FAZs also contain iron oxides that contribute to additional CO_2_ uptake mechanisms [26]. 

The great application potential of zeolite materials has not been fully explored, despite the fact that synthetic zeolites are broadly studied as adsorbents, molecular sieves, ion-exchangers and catalysts for industrial and environmental protection applications due to their unique surface properties [27]. The processing of zeolite thin film and membranes will reveal novel opportunities for their advanced applications. Experimental studies on the deposition of pure zeolite thin films with controllable phase composition have been performed by spin casting or nozzle spray of colloidal suspensions [28,29], in situ crystallization [30], and dry gel conversion [31]. Zeolite membranes are distinguished by unique adsorption and separation properties combined with high selectivity due to their ordered micro-porosity in the structural framework [32]. Zeolite thin films are promising sensing media because of their adsorption properties and ability to host molecules of sensing substances [33]. The embedding of metal nanoparticles in mesoporous oxide thin films results in novel nanocomposite materials with advanced optical, catalytic, and sensing properties derived from the structural synergy of the matrix and the hosted active centers [34]. As the FA is, at the same time, a source of silica, alumina and iron oxides, it could be expected that this initial chemical composition is a prerequisite for the deposition of thin layers of self-assembled composites of zeolite-like matrixes with meso-porous structures, including distributed nanoparticles of iron oxides. At present, the scientific literature on deposition and studies of FAZ thin films is limited to initial investigations on coatings consisting of a sol-gel Nb_2_O_5_ matrix doped with fly ash zeolites [35].

The present paper summarizes our best attempts at the synthesis of zeolites of Na-X type from class F lignite coal fly ash and outlines their possible application in environmental protection systems as adsorbents of carbon dioxide and catalysts for the total oxidation of VOCs. This study complements the existing knowledge, with the comparison of coal ash zeolites Na-X obtained through various synthesis and homogenization techniques, which are reflected in their compositional and textural characteristics. The present study compares the influence of magnetic homogenization and ultrasonic activation of the reaction mixtures on the properties of coal ash zeolites in relation to their use as adsorbents and catalysts. This is the first demonstration of the in situ deposition of thin film of coal ash zeolites and their potential for advanced applications as sensitive media. 

## 2. Materials and Methods 

### 2.1. Characterization of the Raw Material

Fly ash generated by the combustion of lignite coal from the largest coal field located in the area of Radnevo, Bulgaria, “Maritsa East”, was used as a raw material for the synthesis of zeolites. FA was sampled from the electrostatic precipitators of the three main local Thermal Power Plants supplied with lignite, namely TPP “Maritza East 2” (FA_ME_), TPP “AES Galabovo” (FA_AES_,) and TPP “Contour Global” (FA_CG_), which are three of the four main power plants in south-central Bulgaria’s Maritsa East Energy Complex. The chemical composition of different FA samples was examined by common chemical analyses and atomic absorption spectroscopy, as described in [36]. The phase composition of FA samples was studied by X-ray diffraction analyses. Their density was measured pycnometrically, and their granulometry was determined by sieve analysis.

### 2.2. Synthesis and Characterization of Coal Fly Ash Zeolites 

Powdery coal fly ash zeolites were synthesized by alkaline activation of lignite coal FA using three different laboratory procedures: hydrothermal activation (H), double-stage fusion–hydrothermal activation (FH) and atmospheric crystallization (AA). In the hydrothermal activation, suspensions of 5 g FA in 100 mL of sodium hydroxide solutions with concentrations of 2.5 mol/L in closed vessels were heated to 90 °C and kept at this temperature for 2 h. In the two-step fusion–hydrothermal synthesis, a preliminary alkaline melting step was carried out, in which solid phase mixtures of 5 g FA and 10 g sodium hydroxide were heated in nickel crucibles to 550 °C for 1 h. The obtained sintered mixtures were diluted in distilled water and subsequently subjected to hydrothermal activation. In both procedures, the reaction slurries were homogenized prior to the hydrothermal treatment by continuous magnetic stirring for 5 h (M mode) or by ultrasonic treatment (U mode) for 15 min. The applied temperatures and durations of fusion and hydrothermal treatment were found to be optimal for the synthesis of zeolite Na-X in our previous studies [25]. It has been observed that zeolite Na-X is crystallized from the reaction slurries of coal fly ash and alkaline solutions as a metastable phase, and its preparation requires moderate synthesis temperatures and short hydrothermal activation [36]. The optimal duration of magnetic homogenization and ultrasonic treatment of the reaction mixtures for the synthesis of zeolite Na-X has been established experimentally [37]. The homogenization of the mixtures accelerated the release of the aluminosilicate components from the raw FA into the reaction solution in the form of soluble sodium aluminate and sodium silicate. In the third zeolitization approach, slurries of 10 g FA in 100 mL of 1.5 mol/L NaOH solutions in closed polypropylene containers were kept at room temperature over the course of 10 months. The laboratory procedures for the conversion of FA into zeolites are presented in Figure 1.

The obtained FAZ powders were separated by filtration of the reactant solutions, washed with distilled water and dried in an oven at 105 °C. The phase composition of the synthesized FAZs was studied by X-ray diffraction using a Brucker D2 Phaser diffractometer (Bucker Corporation, Germany,) with CuKα radiation and an Ni filter. Their morphology was observed by scanning electron microscopy (SEM) using a Carl Zeiss SMT SEM *EVO* LS25 (Carl Zaiss AG, Germany) with an *EDAX* Trident system. The surface properties of FAZs were investigated via nitrogen adsorption/desorption isotherms measured at liquid nitrogen temperatures using a volumetric adsorption analyzer Tristar II 3020 (Micromeritics Instrument Corporation, USA. Brunauer–Emmett–Teller (BET)-specific surface area (S_BET_, m^2^/g) was evaluated applying the multi-point Brunauer–Emmett–Teller (BET) model to the experimental adsorption data. The mesopore size was calculated by the Barrett–Joyner–Halenda (BJH) model using data from the desorption branch of the isotherms. The micropore volume was evaluated by applying a t-plot model.

### 2.3. Deposition and Characterization of Fly Ash Zeolite Thin Films

Fly ash zeolite thin films (FAZTFs) were deposited onto optical glass substrates by in situ hydrothermal crystallization. The substrates were seeded ultrasonically to create centers of nucleation and were then immersed into the zeolite growth solution. The hydrothermal deposition experiments were performed at 90 °C with varying durations from 4 to 20 h. The resultant coatings were inspected for surface morphology using a Jeol Superprobe 733 scanning electron microscope (JEOL, Tokyo, Japan) and for surface roughness using a Zygo Optical Profiler (Zygo Corporation, Middlefield, CT, USA) at a standard magnification of 50×. The optical transmittance and reflectance spectra were recorded with a high-precision Cary 5E spectrophotometer (Varian, Australia) at a normal light incidence in the wavelength region λ = 200–1200 nm, with an accuracy of 0.1% and 0.3%, respectively. The optical response towards VOCs was studied by reflectance measurements before and after exposure to vapors of acetone selected as probe molecules. 

### 2.4. Investigation of Potential Applications of FAZ

#### 2.4.1. CO_2_ Adsorption 

The adsorption of CO_2_ onto FAZs was studied under equilibrium and dynamic conditions. The equilibrium adsorption potential was measured using Micromeritics equipment at 0 °C in the relative pressure range of p/p_0_ = 0.001–0.030, where p_0_ is the saturation pressure of CO_2_ (3485.6769 kPa, 0 °C). The CO_2_ adsorption on FAZs in dynamic conditions was examined in an adsorption reactor, passing a gas mixture of 10 vol.% CO_2_/N_2_ at a flow rate of 30 mL/min through the adsorbent-fixed bed. Prior to the adsorption tests, the samples were pre-treated in situ for 1 h in nitrogen flow at 400 °C. The effluent gas was analyzed online by GC TCD with a 25 m PLOT Q and HP-5MS capillary columns.

#### 2.4.2. Catalytic Oxidation of VOCs

Parent FAZs or modified FAZs with CuO (Cu–FAZs) were studied in total oxidation of VOCs. Cu–FAZs were obtained by an incipient wetness impregnation technique of the parent FAZs with Cu(NO_3_)_2_.3H_2_O. The copper salt was dissolved in distilled water in an amount corresponding to 5 wt.% of Cu loading, and mixed with FAZ materials. The catalysts were dried at ambient temperature and the salt precursor was decomposed at 400 °C. The catalytic tests were performed using a fixed-bed flow reactor in the temperature range 250–550 °C. All gas lines of the apparatus were heated to 110 °C in order to avoid VOC adsorption on the tube walls. Air used as a carrier gas with a flow rate of 30 mL/min passed through a saturator filled with toluene (the model VOC) and fed into the reactor. A thermocouple was positioned in the catalyst bed for the accurate measurement of the catalyst temperature. The reaction steady state was established after 30 min at each temperature, until the only registered final products of VOC oxidation were CO_2_ and H_2_O. An online analysis of the reaction products was performed using HP-GC (FID/TCD) with a 25 m PLOT Q and HP-5MS capillary columns [12]. 

Prior to the catalytic tests, the samples were pretreated for 1 h in nitrogen flow at 400 °C.

## 3. Results

### 3.1. Characterization of Lignite Coal Fly Ash

The chemical composition, density and granulometry of the FA generated from the lignite coal combusted in the three separate TPPs varied in the intervals listed in Table 1. The content of the main components in the FA compositions varies within ±3 wt.% of the average, depending on the incineration system. The density and granulometry of FA exhibit larger variations, most likely due to the specifics of the coal-grinding systems in the individual TPPs. FA from lignite coal from the “Maritza East” basin is attributed to class F according to the international crystallographic standard ASTM C618, because the total content of SiO_2_ + Al_2_O_3_ + Fe_2_O_3_ was above 70 wt.%. CaO was under 20 wt.%, regardless of the TPP in which the FA was generated. 

A typical X-ray diffraction pattern of raw FA is plotted in Figure 2. The main crystalline phases of all FA samples are quartz (α-SiO_2_), mullite (3Al_2_O_3_·2SiO_2_), hematite (α-Fe_2_O_3_) and magnetite (γ-Fe_3_O_4_). A slight difference in Ca-containing phases is established in the phase composition of the samples collected from the different TPPs. Calcium oxide is sulfated to gypsum (CaSO_4_) in FA_ME_, while in FA_AES_ and FA_CG_ it is included in the aluminosilicate-phase anorthite (CaAl_2_Si_2_O_8_). However, the ratio of amorphous to crystalline constituents in the different FAs varies from 0.8 to 2.0 depending on the combustion systems.

### 3.2. Coal Fly Ash Zeolites

The synthesis conditions and the sample notations are given in Table 2. X-ray diffractograms of FAZs obtained by different synthesis procedures are presented in Figure 2 in comparison to a diffraction pattern of a reference zeolite Na-X. The investigations performed on the lignite coal ash zeolitization reveal that the main zeolite phase, crystallized by three conversion approaches applied at the settled alkaline treatment conditions, is zeolite Na-X. The raw FA is highly silicically obeyed by the lignite coal chemistry, and it is a suitable starting material for the synthesis of zeolites of faujasite type, regardless of the source of its generation. The variation in the ash composition in the ranges indicated in Table 1 does not affect the type of zeolite phase obtained. 

Generally, the synthesis of zeolite Na-X requires mild hydrothermal treatment temperatures (90 °C), a short synthesis duration (2–4 h) and soft alkaline activator molarities (2.5 mol/L), as it appears as a metastable zeolitic phase. Hydrothermal activation results in the low crystallinity of the zeolitization product when synthesizing defined zeolite phases. In general, an increase in zeolite yield is achieved if the product is a mixture of zeolite phases [37]. The alkaline resistant aluminosilicate phases in the raw FA, such as quartz, mullite and anorthite, are also observed in FAZs obtained by hydrothermal synthesis (samples FAZ1 and FAZ2), indicating that they are not utilized in this processing procedure (Figure 2).

The mild temperatures required for the crystallization of zeolite Na-X point out the possibility that this material can be obtained by converting the raw FA under room temperature, at the expense of longer processing times. The results reveal that the synthesis of Na-X FAZs via atmospheric aging requires the crystallization of FA in 1.5 mol/L NaOH over ten months. Despite the technological difficulties posed by the long conversion process to obtain a satisfactory degree of crystallization in the product, this technological variant for the extraction of FAU should not be neglected because of its benefits—zero-energy demands, low alkaline agent consumption and simple equipment. The conversion of FA to zeolites by atmospheric crystallization is feasible even at landfill sites. In the FAZs obtained by atmospheric crystallization, the alkaline-resistant phases in the raw material also remain unconverted, as their characteristic reflections are observed in the experimental X-ray diffractograms (Figure 2, sample FAZ 5). In the case of double-stage fusion–hydrothermal synthesis, the stable aluminosilicates in the FA are assimilated in the zeolitization process due to the high temperature treatment of the reaction mixture at 550 °C for 1 h prior to the hydrothermal activation. This was confirmed by the absence of their characteristic reflections in the X-ray diffractograms (Figure 2, sample FAZ 3). X-ray diffraction analyses reveal intensive lines of zeolite Na-X, indicating the strong crystallization of this zeolite phase in the samples obtained by double-stage syntheses. 

The most important factors for the process of zeolitization of FA are the content, ratio and phase state of the Si- and Al-components in the composition of the raw material. Regarding its aluminosilicate components, the FA from the three TPPs is suitable for the synthesis of high-silica zeolites of the fuajasite type that require optimal molar ratios of SiO_2_/Al_2_O_3_ above three [10]. The higher amorphous part in FA facilitates the zeolitization process because of the better solubility of the amorphous aluminosilicates in the alkaline media. In contrast to this, the higher content of alkaline-resistant phases such as quartz (α-SiO_2_) and mullite (Al_6_Si_2_O_13_) hinders the zeolitization. The dissolution of the crystalline aluminosilicates in the alkaline media is accelerated by the preliminary fusion stage. 

The findings show that iron oxides do not influence the zeolitization, but are incorporated into the resulting zeolite structure, appearing as seeds around which zeolite crystals grow, or as nanoparticles incorporated into the cages and channels of the zeolite structures. Ferrous and ferric ions released by the alkaline dissolution of the amorphous constituents of the FA can also take part as compensating cations in the zeolite frameworks. The synthesis of FAZs via hydrothermal activation at moderate temperatures ensures the transfer of spinel and non-spinel ferrous oxides into the reaction products [38]. High-temperature alkaline melting leads to the oxidation of spinel iron oxides, resulting in hematite formation [39].

The homogenization of the reaction mixtures prior to the hydrothermal treatment was carried out using two different methods: magnetic stirring or ultrasonication. Figure 3 presents the SEM micrographs of the raw FA and FAZs of Na-X type obtained by atmospheric crystallization and double-stage fusion–hydrothermal syntheses with continuous magnetic stirring or short ultrasonic treatment. 

FA is composed of species of micron sizes with different morphologies (Figure 3a). The crystallites of the zeolite Na-X exhibit various morphologies depending on the synthesis and the homogenization procedures applied. Samples obtained by atmospheric crystallization are composed of agglomerates (Figure 3b), while the FAZs crystallized from the preliminary magnetically or ultrasonically homogenized slurries are composed of individual crystals (Figure 3c,d). Ultrasonic treatment of the reaction mixtures results in nanocrystalline synthesis products (Figure 3d), while the magnetic stirring leads to the formation of micron-sized crystallites (Figure 3c). 

Table 2 presents the results for the main surface parameters of selected FAZs of Na-X zeolite type obtained by different synthesis modes. The surface characteristics of a reference zeolite Na-X synthesized from pure starting materials in a stoichiometric ratio are presented for comparison. Typical experimentally measured N_2_ adsorption/desorption isotherms of FAZs are plotted in Figure 4. The figure reveals the different adsorption stages: (i) fast filling in micropores at low pressure ratios p/p_0_, (ii) the region of monolayer formation (isotherm’s knee), and (iii) continued adsorption in mesopores at p/p_0_ above 0.003. Well-expressed hysteresis loops are exhibited between the adsorption and desorption branches of the isotherms for all investigated FAZs. The values of BET-specific surface areas (S_BET_, m^2^/g), external surface areas (S_extern_, m^2^/g), total (V_total_, m^3^/g) and micro pore volumes (V_micro_, m^3^/g), diameters of micro- (d_micro_, Å) and mesopores (d_meso_, Å) were calculated by model studies of the isotherm data (Table 2). The S_BET_ of FAZs varies in the range of 73–486 m^2^/g. The lowest specific surface value was measured for the FAZ synthesized by hydrothermal treatment, while the highest S_BET_ value was achieved for FAZ3 obtained by ultrasonically assisted double-stage fusion–hydrothermal synthesis. The ultrasonic homogenization contributes to the expansion of the specific surface area for both two-stage and hydrothermal syntheses.

The S_BET_ values of coal ash zeolites remain lower in comparison with those of the pure synthetic zeolite Na-X. FAZs are characterized by a mixed micro–mesoporous structure, as revealed by the experimental N_2_ adsoption/desorption and model porosity studies. N_2_ isotherms of FAZs can be assigned to type IV, according to the International Union of Pure and Applied Chemistry (IUPAC) classification (1985), with broad hysteresis loops of the H3 type typical of micro–mesoporous textural materials. The values of the volume described by micropores V_micro_ (m^3^/g) are higher for the ultrasonicated FAZs than in the cases when magnetic stirring was applied, which indicates that ultrasonic treatment results in a higher share of microporosity (Table 2). 

The zeolitization extent Z for FAZs of Na-X type is calculated on the basis of the S_BET_ values reported in Table 2 and _by_ applying the following equation:(1)$$Z=\frac{S_{BET,\exp}}{S_{BET,ref}}\frac{1}{\left(SiO_{2}+Al_{2}O_{3}\right)}100,wt.\%$$
where Z is the yield of the zeolite Na-X phase from the aluminosilicate part of the raw FA, wt.%, S_BET,exp_ and S_BET,ref_ are the BET specific surface areas for FAZ and the referent zeolite Na-X, correspondingly and SiO_2_ and Al_2_O_3_ are the contents of the corresponding components in the FA composition, wt.%/100. 

The evaluation of zeolitization extent based on specific surface values is one of the reliable methods applied, as discussed by Majchrzak-Kuceba [40].

A higher zeolitization extent is achieved at higher amorphous to crystalline ratios in the raw FA due to the faster alkaline dissolution of the amorphous aluminosilicates when compared to the crystalline phases. A maximal Na-X zeolite yield of 89 wt.% from the aluminosilicate part of the raw material was achieved by applying a double-stage fusion–hydrothermal synthesis in combination with ultrasonic treatment. 

### 3.3. Fly Ash Zeolite Thin Films

Typical SEM micrographs and optical profilograms of fly ash zeolite thin films (FAZTFs) are presented in Figure 5. The morphological studies reveal the formation of a continuous layer after at least 8 h of deposition. A smooth and uniform surface was obtained for the FAZTF2 after 12 h of crystallization. The longer deposition period results in a higher roughness and a more granular microstructure. The optical transmittance and reflectance spectra of FAZTFs obtained for different deposition times are plotted in Figure 6a,b, respectively. The films are characterized by high transmittance in the visible and near IR (VIS-NIR) spectral ranges; the transmission coefficient values decrease after a longer crystallization duration. 

Figure 6c presents the relative change in the reflectance after exposure of FAZTFs to acetone vapors. When porous films are exposed to acetone vapors, condensation takes place in the pores. As a result, the effective refractive index of the films increases because air (refractive index of 1) is replaced by a material with a higher refractive index (i.e., liquid acetone). Since the film reflectance is a function of the film optical constants, a change in the reflectance spectra is observed as well. Figure 6c shows that the reflectance response is the highest for the film FAZTF2 obtained after 12 hours of crystallization and that FAZTF2 possesses the smoothest surface.

### 3.4. Adsorption Potential of FAZs for CO_2_ Capture

Equilibrium adsorption isotherms of CO_2_ measured onto FAZs at 0 °C and pressure up to 105 kPa are plotted in Figure 7a. Sonicated FAZs have lower adsorption capacity for CO_2_ relative to unit surface when compared to FAZs synthesized by the magnetic stirring of the reaction mixtures. Thus, despite its lower surface value, FAZ4 possesses a similar and even higher ability to capture CO_2_ compared to FAZ3; FAZ5 exhibits significantly higher adsorption capacity than FAZ2, despite their close S_BET_ values. This trend can be explained by the differences in the external surface values (S_extern_, m^2^/g) of the zeolite particles synthesized in different homogenization modes (Table 2). Ultrasonic treatment results in a higher share of S_extern_ in the total S_BET_ value due to the smaller particle size of FAZs obtained by ultrasound-assisted syntheses. This leads to a lower adsorption into the pores of the zeolite framework. The iron oxides incorporated into the FAZ structural frameworks influence the CO_2_ adsorption capacity of the synthesized materials. Previous studies have shown that the iron oxide doping of zeolites increases CO_2_ adsorption due to the presence of a higher amount of coordinately unsaturated iron, creating various adsorption sites. These sites provoke a stronger interaction with CO_2_ molecules due to higher unsaturation on the solid surface and a stronger interaction between the adsorbent and the gas molecules [41]. In our previous studies, it was found that the sonicated FAZs obtained by a double-stage synthesis contain mostly ionic iron, which participates as a compensating cation in the zeolite framework compared to the zeolite samples obtained by magnetic stirring, in which iron oxides are predominately formed [12]. 

The breakthrough curves of dynamic CO_2_ adsorption onto FAZ3 and FAZ4 at different temperatures of CO_2_ desorption are plotted in Figure 7b. The findings show that the CO_2_ adsorption process is faster for sonicated FAZs. This could also be explained by their higher external surface area, which ensures a shorter time for mass transfer.

Synthetic faujasite and linde are considered the most promising zeolite adsorbents for carbon capture technologies due to the combination of their low-cost processing, large surface areas, acceptable adsorption potential of CO_2_ at ambient temperatures and atmospheric pressures, and high CO_2_/N_2_ separation capacity [42]. The disadvantages that limit their broad application for carbon capture include the high energy demands for their thermal recovery and the preferential adsorption of moisture. However, CO_2_ adsoption/desoption studies on FAZs demonstrate a favorable CO_2_ desorption process due to the textural peculiarities of these materials. Unlike the pure synthetic zeolites that are microporous, FAZs are distinguished with a mixed micro–mesoporous framework that facilitates mass transport phenomena through the material and favors the desorption process. Figure 7b demonstrates the thermal recovery of the sample FAZ4. Breakthrough curves of dynamic CO_2_ adsortion onto the zeolite sample recovered at 120 °C and 60 °C are plotted. It is observed that FAZ4 retains its high adsorption capacity even at the lower regeneration temperature. For comparison, a lower recovery rate of microporous zeolites by thermal CO_2_ desorption was achieved at 300 °C, as reported by Lee and Park [42]. This profitable thermal recovery of coal fly ash zeolites compared to their pure synthetic microporous analogues is obeyed by their mixed micro–mesoporous structure. FAZs possess a CO_2_ capture ability comparable to that of pure synthetic zeolites, in addition to a lower cost and favorable regeneration [17]. 

### 3.5. Catalytic Activity of FAZs for Total Oxidation of VOCs

The catalytic performance of FAZ3 and FAZ4 samples in the total oxidation of toluene is presented in Figure 8. These samples were selected because of their high surface area and the presence of finely distributed iron oxide particles, which can contribute active sites in the catalytic process. The sonicated sample FAZ3 exhibits stronger catalytic activity in the studied temperature range when compared to the sample obtained under magnetic stirring (Figure 8). 

Modification with CuO significantly improves the catalytic ability of FAZ4 synthesized in M-mode, despite the reduction in the surface parameters due to the loading with metal oxide nanoparticles (see Table 2 and Figure 8). A toluene conversion rate of 50% was achieved at 300 °C for CuO-modified FAZ and at 480 °C for the parent FAZ. Reflections typical for CuO crystallites are observed in the experimental X-ray diffractograms for Cu-modified FAZ (Figure 2, Cu–FAZ4). The metal particles fill the micropores of the zeolite structure, reducing the internal volume determined by the micropores (V_micro_) of the FAZs (Table 2). The reduction in the specific surface area of the modified samples is more pronounced for sonicated FAZs, as they yield more micropores (Table 2). The modification with metal oxide particles leads to a decrease in the surface area and pore volume of the Cu–FAZ materials due to the pores filling with metal oxide species, but the distorting effects on the zeolitre structure due to the modification process cannot be excluded, as it was indicated by X-ray diffraction (Figure 2).

We demonstrate that the catalytic activity of FAZs in the total oxidation of toluene depends on the specific surface of the zeolite and the dispersion of iron oxide particles. The results of the catalytic activity of FAZs reveal an interaction between the metal oxide particles and the carrier that increases the number of active reaction sites on the surface of the catalysts. Due to the ultrasonic treatment, the sonicated FAZs are characterized by their nano-sized crystalline morphology, higher concentration and finer distribution of iron oxides, which serve as active centers for the easier release of oxygen atoms for the oxidation of VOCs. In contrast to the microporous zeolites, the mixed micro–mesoporous structure of coal ash zeolites improves their catalytic activity due to the accelerated transport of reagents across the material. The obtaining of hierarchical FAZs by modification with CuO particles significantly improves their catalytic performance due to the formation of additional active centers that facilitate the formation of oxygen defects (oxygen vacancies) on the catalyst surface, which are needed for the reaction process. The combination of the high catalytic activity of the modified FAZs in the total oxidation of VOCs (to CO_2_ and water) with the high adsorption potential for carbon emission capture of the initial FAZs reveals an opportunity for the development of a zero-emission VOC combustion technology based on solid waste utilization, which would provide significant environmental benefits. 

## 4. Conclusions

Coal fly ash zeolites of Na-X type were obtained from coal ash with variable aluminosilicate content (within ±3 wt.%) using three zeolitization techniques: atmospheric crystallization, hydrothermal activation, and double-stage fusion–hydrothermal activation. Depending on the homogenization of the reaction mixtures, magnetic stirring or ultrasonic treatment, coal ash zeolites of micro- and nanocrystalline morphology were obtained. A zeolitization extent of 89 wt.% of the starting aluminosilicate composition can be achieved, with specific surface values of the resulting zeolite Na-X above 480 m^2^/g, as demonstrated for the case of double-stage fusion–hydrothermal synthesis with ultrasonic treatment. The surface studies of the FAZs reveal a mixed micro–mesoporous texture that facilitates mass transport phenomena through the materials. Coal fly ash zeolites with a specific surface area of about 400 m^2^/g exhibit high CO_2_ adsorption capacity above 3.0 mmol/g FAZ under atmospheric pressure. The adsorption of CO_2_ under dynamic conditions proceeds faster with the zeolites synthesized by ultrasonic treatment, but when the magnetic stirring was used, a higher adsorption capacity per surface area of the zeolite was established. Due to the structural peculiarities, FAZs surpass their pure synthetic microporous analogues with lower energy demands for thermal recovery. A high regeneration rate was found at 60 °C. Sonicated coal ash zeolites exhibit a stronger catalytic activity for the total oxidation of toluene above 75% at 490 °C compared to magnetically treated ones. The finer distribution of Fe^2+^/Fe^3+^-active centers in FAZ frameworks achieved by sonication leads to higher catalytic activity in the total oxidation of VOCs. A post-synthesis modification with the copper oxide of FAZs obtained under magnetic stirring significantly improves their catalytic activity due to the formation of additional catalytic sites. Copper-modified FAZs exhibit the total oxidation of toluene above 90% at 400 °C. Coal ash zeolite thin films with uniform morphologies were deposited in situ onto optical glass substrates via the hydrothermal activation of alkaline coal ash slurries. An optical characterization of the obtained coal ash zeolite layers revealed promising results for further investigations of these materials as optical sensitive media. A significant change in the reflection coefficient was detected by the exposure of thin layers in acetone vapors.

Zeolites obtained from coal ash residues are promising candidates for applications in environmental protection systems for CO_2_ capture, catalytic degradation of volatile organic pollutants and the optical detection of air pollutants due to the unique combination of their particular morphology, composition, surface characteristics and low-cost processing. Studies will continue in the following directions: an investigation into the catalytic behavior of coal ash zeolites in mixtures of VOCs; an evaluation of the selectivity of CO_2_ adsorption in the presence of associated gases; model studies of the adsorption process; and the development of host–guest modifications of thin layers to increase their optical sensitivity. The use of the abundant coal ash resources in gas-cleaning technologies is a big step toward the realization of zero-emission power plants.

## Figures and Tables

**Figure 1 materials-13-02014-f001:**
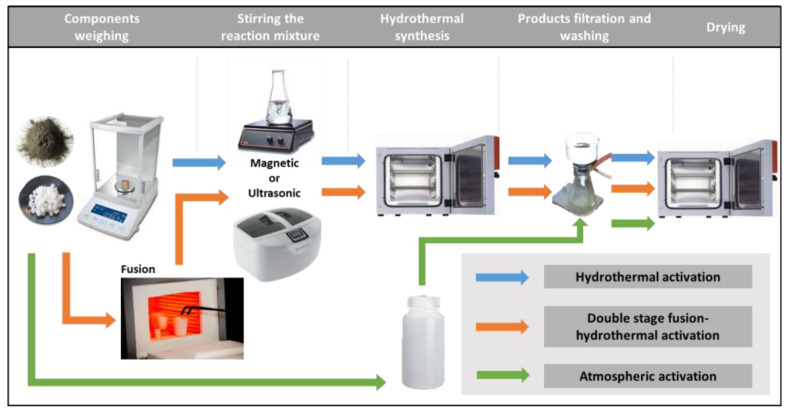
Synthesis of fly ash zeolites by hydrothermal activation, double-stage fusion–hydrothermal activation and atmospheric crystallization.

**Figure 2 materials-13-02014-f002:**
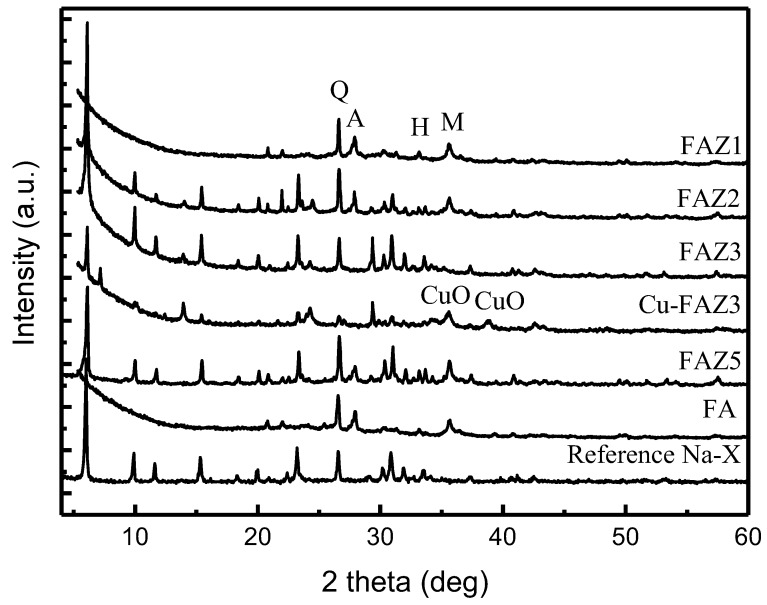
X-ray diffraction patterns of raw FA, reference Na-X, FAZs obtained by hydrothermal activation (FAZ1, FAZ2), double-stage fusion–hydrothermal synthesis (FAZ3) and atmospheric crystallization (FAZ5), and CuO-modified FAZ4 (Cu–FAZ4) (quartz (Q), anorthite (A), hematite (H), magnetite (M)).

**Figure 3 materials-13-02014-f003:**
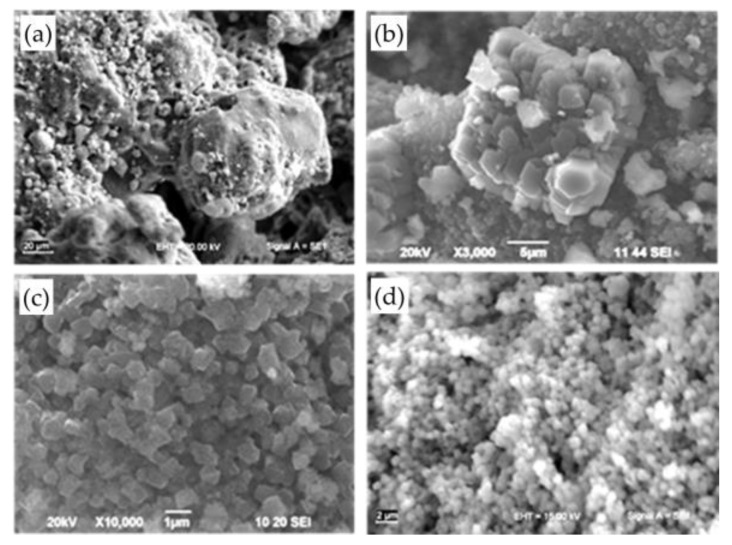
SEM images of raw FA (**a**) and FAZs synthesized by atmospheric crystallization (**b**), fusion–hydrothermal synthesis with magnetic stirring (**c**), and ultrasonic homogenization (**d**).

**Figure 4 materials-13-02014-f004:**
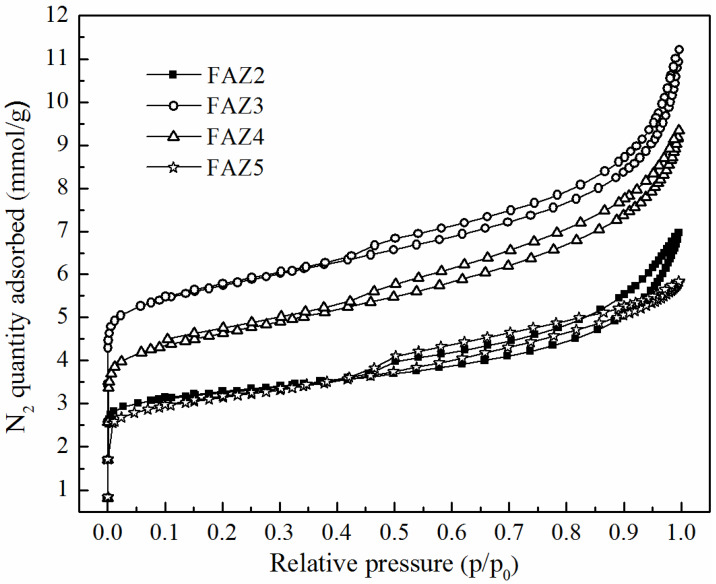
Experimental N_2_ adsorption/desorption isotherms of FAZs.

**Figure 5 materials-13-02014-f005:**
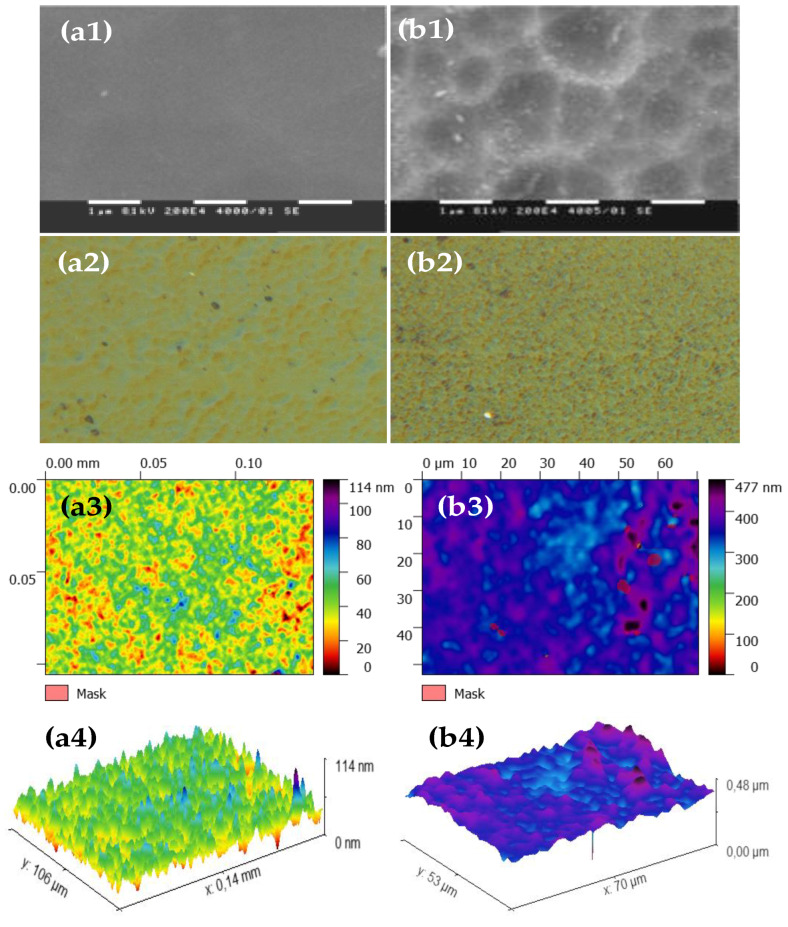
SEM micrographs (**a1**), optical microscope images (**a2**) and profilograms (top view (**a3**) and profile view (**a4**)) of fly ash zeolite thin films (FAZTFs) obtained at different crystallization durations: (**a**) hydrothermal activation for 12 hours; (**b**) hydrothermal activation for 20 hours.

**Figure 6 materials-13-02014-f006:**
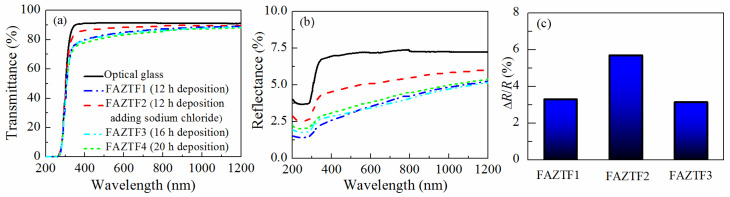
Optical transmittance (**a**) and reflectance (**b**) spectra of FAZTFs deposited at various crystallization times, and reflectance response (dR/R, %) of FAZTFs exposed to acetone vapors (**c**).

**Figure 7 materials-13-02014-f007:**
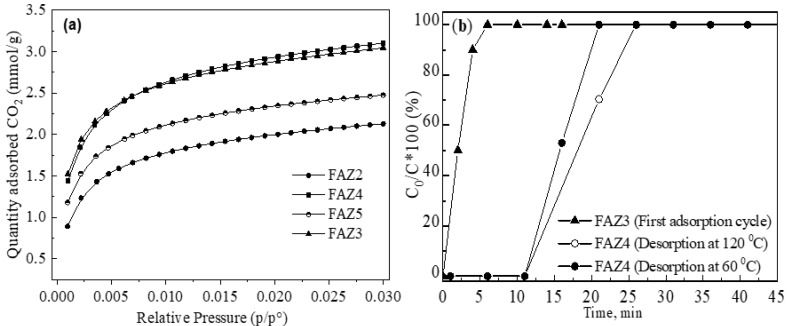
CO_2_ adsorption studies: equilibrium isotherms (**a**) and dynamic breakthrough curves (**b**).

**Figure 8 materials-13-02014-f008:**
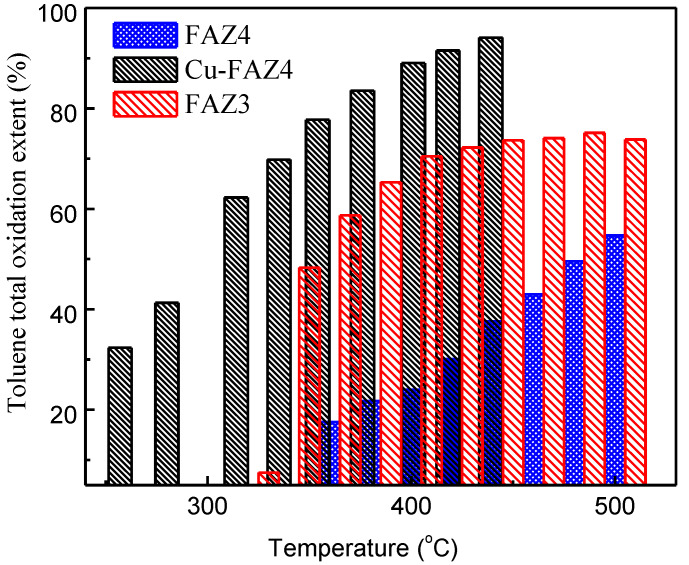
Toluene conversion extent at different temperatures of sonicated FAZ3, parent FAZ4 and CuO-modified FAZ4 obtained with magnetic stirring.

**Table 1 materials-13-02014-t001:** Variation in the chemical composition and some physicochemical properties of lignite coal fly ash (FA) sampled from different Thermal Power Plants (TPPs).

Components	SiO_2_	Al_2_O_3_	Fe_2_O_3_	MgO	SO_3_	CaO	Others
Content, wt.%	50 ± 3	24 ± 1	12 ± 3	2 ± 1	2.25 ± 0.25	4.5 ± 1.5	< 1
Si to Al molar ratio	3.3–3.8						
Density, g/cm^3^	1.9–3.1						
Granulometry, µm	45–250						

**Table 2 materials-13-02014-t002:** Surface parameters and zeolitization extent of fly ash zelolites (FAZs), CuO-modified FAZs and a referent zeolite Na-X.

Sample	Synthesis Mode *	Homoge-Nization **	S_BET_, m^2^/g	S_extern_ m^2^/g	V_total_, m^3^/g	V_micro_, m^3^/g	d_micro_, Å	d_meso_, Å	Zeolitization Extent, wt.%
FAZ1	H	M	73	67	0.11	0.02	11.03	50.17	13.35
FAZ2	H	U	280	87	0.21	0.08	13.81	49.98	51.40
FAZ3	FH	U	486	166	0.31	0.13	13.94	41.79	89.22
FAZ4	FH	M	396	125	0.26	0.11	13.75	44.89	72.65
FAZ5	AA	-	283	98	0.20	0.07	13.65	43.33	49.60
Cu–FAZ3	FH	U	67	47	0.08	0.009	12.54	51.02	-
Cu–FAZ4	FH	M	224	76	0.16	0.06	13.76	46.07	-
Na-X	H	M	780	61	0.33	0.28	13.49	35.26	100.00

* Hydrothermal activation (H); atmospheric crystallization (AA); double-stage fusion–hydrothermal synthesis (FH); ** magnetic stirring (M); ultrasonic treatment (U).

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
