# Peer review of "Progress in the Utilization of Coal Fly Ash by Conversion to Zeolites with Green Energy Applications"

_materials, 2020, doi:10.3390/ma13092014_

Round 1
Reviewer 1 Report
The article is of potential interest. I would like to thank authors for there efforts. My comments are as: 1. The article seems more like review paper. Can you elaborate more about the main objectives of the research, why was the need of the current study and its need, or its contribution to the study area. 2. section 2.1. I recommend presenting a table with the chemical properties of sample material FA used in this study. section 2.2. Briefly mention the rational for selecting temperature and time duration as 90 degree, 2.5 hrs, 1 hr, 5 hrs and 15 min (line 154-160) 3. line 2011: "The effluent gas was analyzed on-line by GC TCD analysis". cite online source 4. results and discussion are presented are fine. 5. conclusion: I recommend authors to work a bit on conclusion section: Include the major finding on FAZs from this study, its applicability and future research need.Author Response
The authors are thankful to the reviewer for the overall positive assessment of our research and the recommendations made to supplement the highlights of the publication.
1) The article seems more like review paper. Can you elaborate more about the main objectives of the research, why was the need of the current study and its need, or its contribution to the study area.
Answer: An additional paragraph was added into the text to clarify the motivation.
2) section 2.1. I recommend presenting a table with the chemical properties of sample material FA used in this study.
Answer: Table 1 contains the chemical composition of the starting coal fly ashes used as raw materials in this study.
3) section 2.2. Briefly mention the rational for selecting temperature and time duration as 90 degree, 2.5 hrs, 1 hr, 5 hrs and 15 min (line 154-160)
Answer: The selected experimental conditions were explained in the revised text.
3) line 2011: "The effluent gas was analyzed on-line by GC TCD analysis". cite online source
Answer: The citation was added.
- Results and discussion are presented are fine.
Answer: The authors are thankful for the positive statement
- conclusion: I recommend authors to work a bit on conclusion section: Include the major finding on FAZs from this study, its applicability and future research need.
Answer: The section conclusions were improved.
Reviewer 2 Report
From an overall point of view, the manuscript is varied and interesting. As said by the authors, most of the countries in the world face the problem of the disposal of fly-ash deriving from power plants that use fossil fuels, trying to reduce the huge CO2 emissions at the same time. In this context, the theme proposed by this manuscript is current and important
In the manuscript, three different hydrothermal synthesis procedures to obtain Na-X zeolites from fly-ash are compared. Subsequently, for these zeolitic materials, uses are proposed as CO2 adsorbents, as sensors of pollutants in the form of thin films and as catalysts for the oxidation of flying organic compounds. The reading of the works already published and the method of exposing the topics in this manuscript, highlight the good familiarity of the authors with the proposed themes. Despite this, I think this work needs some adjustments. I recommend the publication on Materials not before having addressed the following points:
1) In the Abstract, line 29 - "when exposed to various pollutants in the gas phase". Indeed, in paragraph 3.3 the study is related only to acetone.
2) Line 208 - "in the temperature range 250-550 ° C". In figure 8 the conversion of toluene is evaluated up to 755 ° C.
3) Line 225 - the correct standard is ASTM C618.
4) Figure 2 - The XRD graphs should be more distant from each other, within a larger figure. At the same time, I would move the figure in the next paragraph (3.2) to keep it as close as possible to table 2, to have a more immediate confirmation of the names of the various samples.
5) Again with regard to Figure 2 - it is questionable that the Cu-FAZ4 diffractogram is present and not that of its precursor FAZ4.
6) Figure 4 - I would also make this figure bigger because in the printed version it is difficult to read the magnification adopted for the micrographs.
7) Line 340 - I probably did not understand it, but in the bibliographical reference [39] (which compares the estimates on the crystallinity made with the XRD and the IR) I have not found the formula as it is proposed.
8) Line 345 - At the end of the sentence there should be "Wt% / 100".
9) Paragraph 3.4, line 376 -The whole discussion should be slightly revised because:
- a) FAZ3 (the sonicated sample according to table 2) has a surface area greater than FAZ4 so the sequence should be (FAZ2<FAZ5<FAZ4<FAZ3);
- b) line 377 - The proposed explanation that sonicated samples have greater CO2 capture capacity and therefore FAZ3 has an adsorption isotherm similar to that of FAZ4, it does not work because it is precisely the sonicated sample (FAZ3) that gives a result, below expectations.
- c) line 381 "Thus, despite its lower surface value FAZ3 possesses similar ability to capture CO2 as compared to FAZ4" is a further phrase that does not agree with the data previously proposed in Table 2
- d) The adsorption curves of FAZ3 and FAZ4 are practically coincident except for the experimental errors. Are the authors able to explain why 100 m2/g more surface of the sonicated sample have no effect in this test?
10) Figure 7b - On the ordinate axis should go C / C0 · 100 if C0 indicates the initial concentration of CO2. At what temperature was the FAZ3 breakthrough curve obtained?
11) Line 406 - "As seen in figure 7b, higher recovery of FAZs can be achieved at 60 ° C". Could the authors spend a few more words to describe the characteristics of the curves represented and the meaning of the previous sentence?
12) Line 428 - This point is only my opinion and does not want to be coercive with respect to the publication of the work. In any case, I'd like to have an opinion from the authors. Several authors who have dealt with incipient wet impregnation of zeolites with copper affirm in their works concepts similar to the phrase of line 428 "The modification with metal oxide particles does not affect their crystalline structure as confirmed by X-RAY diffraction". In any case, I don't fully agree because looking at the xrd you can notice a certain decrease of the zeolitic crystallinity. Also in this work, although the FAZ4 diffractogram is not present, it seems to me that a not negligible decrease occurred on Cu-FAZ4. The drastic reduction of the surface area of ​​this sample is certainly attributable to the copper oxides that filled the porosity of the material, but in my opinion, not only to them. There could be two further non-negligible effects during the impregnation time:
- a) ion exchange with Cu2+ which can have distorting effects on the structure and which could "hide" the cations from atmospheric oxygen during the heat decomposition treatment of the salt, forcing Cu to bind with oxygen from the matrix.
- b) The rather acidic pH that copper nitrate determines in an average concentrated solution.
For these reasons I do not think it is correct to report in Table 2, the percentages of crystallinity that coincide between Cu-FAZ and FAZ from which they derive.
Author Response
The authors thank the reviewer for the very constructive recommendations and for the overall positive evaluation of the proposed manuscript. All recommendations of the reviewer were considered in the revised version of the manuscript.
Note: (1) In the Abstract, line 29 - "when exposed to various pollutants in the gas phase". Indeed, in paragraph 3.3 the study is related only to acetone.
Answer: The analyte used for the performed tests was specified.
Note: (2) Line 208 - "in the temperature range 250-550 °C". In figure 8 the conversion of toluene is evaluated up to 755 ° C.
Answer: The studied temperature range is 250-550 °C. But in Fig. 8 there was a mistake found in the presentation of the results in different dimensions of temperature (K and Celsius) for different samples. Fig.8 was corrected in the revised version and the related text. We thank the reviewer very much for noticing this inaccuracy.
Note: (3) Line 225 - the correct standard is ASTM C618.
Answer: The number of the standard was corrected in the text.
Note: (4) Figure 2 - The XRD graphs should be more distant from each other, within a larger figure. At the same time, I would move the figure in the next paragraph (3.2) to keep it as close as possible to table 2, to have a more immediate confirmation of the names of the various samples.
Answer: In Figure 2 the individual X-ray patterns were spaced as suggested by the reviewer. In the revised version Fig. 2 was moved in section 3.2.
Note: (5) Again with regard to Figure 2 - it is questionable that the Cu-FAZ4 diffractogram is present and not that of its precursor FAZ4.
Answer: The X-ray pattern of the original FAZ4 reveals an identical phase composition to the modified sample typical for zeolite Na-X, with two additional reflexes attributed to CuO being detected with Cu-FAZ. This was explained in the text. We considered the recommendation to add an X-ray diffractogram of the parent sample, but Figure 2 becomes overwhelmed with data, and this renders the consideration of the recommendation (4) difficult.
Note: 6) Figure 4 - I would also make this figure bigger because in the printed version it is difficult to read the magnification adopted for the micrographs.
Answer: Fig. 4 was enlarged, as recommended by the reviewer.
Note: (7) Line 340 - I probably did not understand it, but in the bibliographical reference [39] (which compares the estimates on the crystallinity made with the XRD and the IR) I have not found the formula as it is proposed.
Answer: The evaluation of zeolitization extent based on specific surface values is one of the reliable methods applied. A comparative examination of the different crystallinity estimation methods was carried out by Majchrzak-Kuceba. Ref [39] was replaced by the appropriate literature source.
Note: 8) Line 345 - At the end of the sentence there should be "Wt% / 100".
Answer: The unit was corrected.
Note: 9) Paragraph 3.4, line 376 -The whole discussion should be slightly revised because:
- a) FAZ3 (the sonicated sample according to table 2) has a surface area greater than FAZ4 so the sequence should be (FAZ2<FAZ5<FAZ4<FAZ3);
- b) line 377 - The proposed explanation that sonicated samples have greater CO2 capture capacity and therefore FAZ3 has an adsorption isotherm similar to that of FAZ4, it does not work because it is precisely the sonicated sample (FAZ3) that gives a result, below expectations.
- c) line 381 "Thus, despite its lower surface value FAZ3 possesses similar ability to capture CO2as compared to FAZ4" is a further phrase that does not agree with the data previously proposed in Table 2
- d) The adsorption curves of FAZ3 and FAZ4 are practically coincident except for the experimental errors. Are the authors able to explain why 100 m2/g more surface of the sonicated sample have no effect in this test?
Answer: We are very thankful to the reviewer for noticing this discrepancy in the discussions. The Paragraph concerning the CO2 adsorption was rewritten in the revised version. Additional experimental data for external surface values were added in Table 2 in support of the analysis of the results.
Note: 10) Figure 7b - On the ordinate axis should go C / C0 · 100 if C0 indicates the initial concentration of CO2. At what temperature was the FAZ3 breakthrough curve obtained?
Answer: Fig 7,b was amended, as suggested.
Note: 11) Line 406 - "As seen in figure 7b, higher recovery of FAZs can be achieved at 60 ° C". Could the authors spend a few more words to describe the characteristics of the curves represented and the meaning of the previous sentence?
Answer: Additional explanations were added in to the text.
Note: 12) Line 428 - This point is only my opinion and does not want to be coercive with respect to the publication of the work. In any case, I'd like to have an opinion from the authors. Several authors who have dealt with incipient wet impregnation of zeolites with copper affirm in their works concepts similar to the phrase of line 428 "The modification with metal oxide particles does not affect their crystalline structure as confirmed by X-RAY diffraction". In any case, I don't fully agree because looking at the xrd you can notice a certain decrease of the zeolitic crystallinity. Also in this work, although the FAZ4 diffractogram is not present, it seems to me that a not negligible decrease occurred on Cu-FAZ4. The drastic reduction of the surface area of ​​this sample is certainly attributable to the copper oxides that filled the porosity of the material, but in my opinion, not only to them. There could be two further non-negligible effects during the impregnation time:
- a) ion exchange with Cu2+which can have distorting effects on the structure and which could "hide" the cations from atmospheric oxygen during the heat decomposition treatment of the salt, forcing Cu to bind with oxygen from the matrix.
- b) The rather acidic pH that copper nitrate determines in an average concentrated solution.
For these reasons I do not think it is correct to report in Table 2, the percentages of crystallinity that coincide between Cu-FAZ and FAZ from which they derive.
Answer: We agree that the presentation of the percentages of crystallinity of Cu-FAZ materials is not appropriate as this parameter was used as proof for the zeolitization process of fly ash. The formation of Cu2+ bonded to the zeolite surface cannot be excluded but we observed the predominant formation of CuO, as XRD showed. Precisely theses CuO particles lead to the decrease of surface area and pore volume of the initial materials. The decrease of the intensity of the XRD reflections is probably due to the decrease of the zeolite part in the CuFAZ material but, we agree with the referee that the zeolite structure is affected as well. The text was modified in this respect.
Reviewer 3 Report
The study addresses a scientifically relevant topic, with a well-established analysis approach and, at the same time, original insights and results.
The paper is well structured and clearly written.
Although the English language is not bad, a revision would be advisable because the manuscript is difficult to follow.
Conclusions should be rewritten to clearly highlight the main findings of the paper.
From these reasons, I think that this manuscript after a minor revision could be accepted to be published by the Materials journal.
Author Response
The authors are thankful to the reviewer for evaluating the originality and good quality of the results, the appropriateness of the research approaches applied. The reviewer's recommendations are considered in the revised version.
Although the English language is not bad, a revision would be advisable because the manuscript is difficult to follow.
Answer: The language was consulted with a native speaker and improved.
Conclusions should be rewritten to clearly highlight the main findings of the paper.
Answer: Conclusions were rewritten as recommended.
Round 2
Reviewer 1 Report
I would like to thank authors for addressing the comments/adding objectives and conclusion section as requested. I believe its ready for publication.Author Response
We would like to thank the reviewer for appreciation and for the useful comments.